# Patients’ Health Literacy in Rehabilitation: Comparison between the Estimation of Patients and Health Care Professionals

**DOI:** 10.3390/ijerph19063522

**Published:** 2022-03-16

**Authors:** Mona Voigt-Barbarowicz, Günter Dietz, Nicole Renken, Ruben Schmöger, Anna Levke Brütt

**Affiliations:** 1Junior Research Group for Rehabilitation Sciences, Department of Health Services Research, University of Oldenburg, 26129 Oldenburg, Germany; ruben.schmoeger@uni-oldenburg.de (R.S.); anna.levke.bruett@uol.de (A.L.B.); 2Clinic for Orthopedic and Rheumatological Rehabilitation, Rehabilitation Centre Bad Zwischenahn, 26160 Bad Zwischenahn, Germany; g.dietz@rehazentrum-am-meer.de (G.D.); n.renken@rehazentrum-am-meer.de (N.R.)

**Keywords:** HCP, rehabilitants, agreement, HLQ, intraclass correlation (ICC), physicians, physiotherapists, social workers, nurses, orthopedic

## Abstract

The term health literacy (HL) comprises the handling of health information and disease-specific and generic self-management skills, especially relevant for patients with chronic conditions. Health care professionals (HCPs) should correctly identify patients’ communication needs and their HL levels. Therefore, the aims of the study were (1) to determine inpatient medical rehabilitation patients’ HL based on self-assessment, (2) to evaluate changes from admission to discharge, (3) to identify HCPs estimation of patients’ HL, and (4) to compare the estimated patient HL by patients and HCPs. A combined cross-sectional and longitudinal study was conducted in an orthopedic rehabilitation center in Germany. The multidimensional Health Literacy Questionnaire (HLQ) was filled in by patients (admission, discharge). An adapted version was administered to HCPs (*n* = 32) in order to assess HL of individual patients. Data from 287 patients were used for the longitudinal analysis, and comparison was based on *n* = 278 cases with at least two HL estimations. The results showed a significant increase in HL in five of nine scales with small effect sizes. Moreover, HCPs mostly provided higher scores than patients, and agreement was poor to fair. Differences between the HL estimation might lead to communication problems, and communication training could be useful.

## 1. Introduction

Health literacy (HL) is defined as “the degree to which individuals can obtain, process, understand, and communicate about health-related information needed to make informed health decisions” (p. 16) [1]. These HL skills enable a person to make health-related decisions. Low HL is a public health challenge throughout Europe [2]; previous research showed that almost 50% of adults in eight European countries have problematic or insufficient HL [3]. Approximately 54% of the German population cannot obtain, understand, and use relevant health information for their health choices [4]. Low HL of persons is associated with higher health care costs and poorer health outcomes [5,6,7,8,9]. Among other underprivileged groups, persons with chronic diseases are characterized by a lower level of HL than the general population [10,11].

HL is relevant in all health care sectors, but it is particularly relevant in medical rehabilitation. Medical rehabilitation is particularly important to make rehabilitants experts in their own chronic illnesses and thus positively influence the course [12]. In Germany, medical rehabilitation is characterized by several special features compared to other countries. This form of rehabilitation is predominantly carried out on an inpatient basis in clinics far from the patient’s home, and the duration of rehabilitation is three weeks. Only approximately every seventh rehabilitation is based on outpatient treatment. In general, rehabilitation aims to help people with chronic diseases, disabled people, and people at risk of disability continue living as independently as possible and support a return to work. Medical rehabilitation can be further subdivided into specialist areas, such as orthopedic, neurological, or psychosomatic rehabilitation. In orthopedic rehabilitation, diseases and restrictions of the musculoskeletal system caused by chronic conditions, degenerative diseases, tumors, or accidents are treated. Other distinguishing features include specialization according to indications and the interdisciplinary composition of the rehabilitation team (e.g., physicians, physiotherapists, social workers, nurses, or psychologists) [13].

Especially for patients with chronic diseases and multimorbidity, professional conversations place high demands on those concerned since complex issues have to be discussed [14,15]. Therefore, health care professionals (HCPs) need to be able to adequately estimate the level of HL of their patients and adapt conversations or interventions to them to optimize their effects. Brief communication trainings for HCPs focusing on clear communication skills and HL principles can already improve patient–HCP communication [16,17,18]. Voigt-Barbarowicz and Brütt [19] described in their systematic review that HCPs had difficulty determining patients’ HL adequately. The current state of studies highlighted that most frequently, the estimation of patient HL by physicians was investigated, and other HCP groups, such as nurses, were only considered in one study. Physiotherapists or social workers did not participate in any study.

Furthermore, all studies were conducted in hospital-based care or primary care settings. There are no studies examining the agreement between patients and various HCPs, such as physicians, nurses, physiotherapists, or social workers, in estimating patients’ HL in a rehabilitation setting. In addition, methodological limitations apply, as one-dimensional questionnaires were predominantly used to assess patients’ HL levels.

Therefore, the overall purpose of this study is to supplement the findings of previous research by examining HL levels as estimated by patients and HCPs. Specific aims of the study are (1) to determine inpatient medical rehabilitation patients’ HL based on self-assessment at admission, (2) to evaluate changes in patients’ self-assessed HL from admission to discharge, (3) to identify HCPs estimation of patients’ HL, and (4) to compare the estimated patient HL by patients and rehabilitation HCPs.

## 2. Materials and Methods

### 2.1. Study Design and Setting

A combined cross-sectional (HCP) and longitudinal (patient) study were conducted from September to December 2020 at a rehabilitation clinic in Germany. The study was presented to patients at admission. After consent was obtained, the patients could fill in the questionnaires immediately on location. HCPs completed the questionnaires in their meeting room after patient contact.

### 2.2. Sampling and Participants

The research team handled the recruitment of patients and HCPs in a clinic for orthopedic and rheumatological rehabilitation. Patients with the following indications were treated in orthopedic rehabilitation: diseases and restrictions of the musculoskeletal system caused by chronic conditions, degenerative diseases, tumors, or accidents. Patients might have additional diagnoses (e.g., a cognitive/psychiatric disorder). Inpatients were included in the study if they were of working age (up to 67 years), were at least 18 years old, and had rehabilitation financed by the German Pension Insurance (especially DRV Oldenburg-Bremen, DRV Braunschweig-Hannover, DRV Bund). Patients were excluded if they could not understand the provided questionnaire although supported by a research assistant—these concerned inpatients with language difficulties. Employees from four HCP groups (physicians, nurses, physiotherapists, social workers) were also recruited for the study.

### 2.3. Measures

#### 2.3.1. Health Literacy

For this study, we chose a multidimensional and well-validated instrument [20,21,22], the Health Literacy Questionnaire (HLQ) [23], to measure patients’ HL levels by patients. The HLQ was developed using a “validity driven” approach. The multidimensional measurement is used for surveys, needs assessment, evaluation, and outcomes assessment as well as for informing service improvement and the development of interventions. The HLQ is available upon request [23]. This questionnaire has been tested in various studies and used in many areas, e.g., for population health surveys [24] or the evaluation of health programs [25]. The HLQ comprises 44 items in nine distinct scales:Feeling understood and supported by health care providers (four items)

(Sample question: “I have at least one healthcare provider who…”);

2.Having sufficient information to manage my health (four items)

(Sample question: “I am sure I have all the information I…”);

3.Actively managing my health (five items)

(Sample question: “I spend quite a lot of time actively managing…”);

4.Social support for health (five items)

(Sample question: “I can get access to several people who…”);

5.Appraisal of health information (five items)

(Sample question: “When I see new information about health, I…”);

6.Ability to actively engage with health care providers (five items)

(Sample question: “Make sure that healthcare providers understand…”);

7.Navigating the health care system (six items)

(Sample question: “Decide which healthcare provider you need…”);

8.Ability to find good health information (five items)

(Sample question: “Find health information from several…”);

9.Understanding health information well enough to know what to do (five items)

(Sample question: “Confidently fill medical forms in the correct…”).

Each scale measures an independent domain of HL. The first five scales contain items with responses ranging from one (strongly disagree) to four (strongly agree). The scales six to nine include items with five responses: 1 = cannot do or always difficult, 2 = usually difficult, 3 = sometimes difficult, 4 = usually easy, and 5 = always easy. A higher score indicates greater ability or more support. There is no total score for the HLQ, and it is recommended to calculate the scores for each scale as the mean average of the items within that scale. In different settings, the HLQ demonstrated Cronbach’s alpha > 0.8 for most scales [23].

In this study, the German version of the HLQ (HLQ-G) was used, which was provided by the questionnaire developers, and a license agreement was concluded. This measurement was translated and culturally adapted to the German context, and the psychometric properties of the HLQ-G were ensured. A nine-factor model replication of the English version was achieved with fit indices and psychometric properties similar to the original HLQ. Cronbach’s alpha was at least 0.77 for all scales [20].

#### 2.3.2. Patients Survey

Patients estimated their HL with a paper-based version of the HLQ-G at two measurement points (T0 and T1), based on 44 items representing all 9 scales. In addition, items regarding age, gender, nationality, living situation, education level, employment status, doctor contacts, diseases, and disability were assessed at T0.

#### 2.3.3. HCPs Survey

HCPs were asked to complete the HLQ-G about their patient from their perceptions of their HL status. Therefore, the items were adapted to the wording of the HCPs. For example, “I feel I have good information…” to “I feel **my patient** has good information…”. In other studies, HLQ was also completed by patients and HCPs [21]. In our study, HCPs estimated patients’ HL using only 4 scales (scales 2, 3, 6, 9) [22] and a total of 19 items (abbreviated HLQ-4HCP in the following). The scales focused on having sufficient information (scale 2), actively managing my health (scale 3), engaging with health care professionals (scale 6), and understanding and reading health information (scale 9) because these are important for the rehabilitation process. HCPs could influence these domains in a short time of stay. A selection of scales shortened the questionnaire and made it easier to use in everyday clinical practice. Furthermore, items regarding age, gender, nationality, professional group, and work experience were assessed once at the beginning of the study. Completing the demographic items was voluntary, so HCPs who did not complete the demographics questionnaire could also estimate patients’ HL in the study.

### 2.4. Data Collection

Study participation (HCPs and patients) was voluntary, and written informed consent was obtained. HCPs were informed before the study started. Patients were sent a short information leaflet before admission to the rehabilitation clinic. The patients were informed in detail about the study in the rehabilitation clinic at admission (T0), and the T0 questionnaire was administered. Five days before their discharge (T1), patients received the T1 questionnaire and were asked to complete and submit it in a sealable envelope before discharge. Patients were usually hospitalized for at least three weeks. Patients were surveyed at two points to determine changes in assessment during a rehabilitation stay. The time between T0 and T1 was at least 16 days. The patients had the opportunity to complete the T1 questionnaire from day 16 (until discharge). After patient informed consent, four HCPs from different professional groups (physicians, nurses, physiotherapists, social workers in each patient) were asked to estimate their patients’ HL by the HLQ-4HCPs after a detailed initial interview.

The matching process between patients and HCP estimation was as follows: Patient questionnaires were prepared with ID codes. A code list, including ID and name, was available in the clinic. HCPs indicated patients’ names on their estimation. To match data, names were replaced by ID codes, and the respective data were deleted from the code list. Therefore, data were anonymized after matching.

### 2.5. Data Analysis

The HLQ-G scale scores and demographic data were analyzed descriptively using SPSS (version 27) [26]. According to the HLQ handbook, missing data by patients and HCPs were imputed using the expectation-maximization algorithm. From scales with 4 to 5 items, missing data were imputed only if no more than two questions were missing. From scales with 6 items, missing data were imputed only if no more than three questions were missing within the scale. If there were more missing data, a scale score was not calculated for that individual scale. For demographic data no further imputations were conducted since missing data for each variable were below 5%.

Descriptive statistics were used to analyze demographic data and HLQ data (of patients and estimations by HCPs). Primary inferential statistical analysis was performed using calculations of measures of correlation and by testing for differences in means in dependent and independent samples. The statistical significance level was set at *p* < 0.05, and the Bonferroni method was used to adjust the *p*-values for pairwise comparisons (*p* < 0.05/9 = 0.0055; corrected for 9 pairwise comparisons). Cohen’s [26] measures described the effect size. To measure the agreement between patients’ HL and HCPs, estimation of their patients’ HL at T0 intraclass correlation (ICC 2, k (average measure), two-way random, absolute agreement) was calculated and reported by Cicchetti (1994) guidelines. The guidelines for interpretation for ICC interrater agreement measure are as follows: less than 0.40—poor agreement; between 0.40 and 0.59—fair agreement; between 0.60 and 0.74—good agreement; and between 0.75 and 1.00—excellent agreement [27].

### 2.6. Ethical Considerations and Trial Registration

The study protocol was submitted to the Ethics Committee of the Medical Faculty of Oldenburg (number: 2020-034) for counseling and is in accordance with the Declaration of Helsinki in its current version (World Medical Association (WMA, Ferney-Voltaire, France, 2013). The works council of the rehabilitation center and the German pension insurance also gave their consent to the study. Written informed consent was obtained based on current data protection regulations. Data security was approved by all institutions involved in data collection. The code list is stored separately from the research data.

The study has been registered in the German Clinical Trials Register (DRKS) (registration number: DRKS00021071).

## 3. Results

### 3.1. Sample Characteristics

The demographic characteristics of the patients are shown in Table 1. A total of 361 patients by T0 and 287 patients by T1 participated. Data from 287 patients were used for the analysis. Patients were mostly 50–59 years old (48.8%); 62.0% were female (*n* = 178); 90.9% indicated German as their mother tongue. The highest level of education was a General Certificate of Secondary Education (GCSE) or equivalent (53.3%), followed by A-level education (41.8%). More than half (53.7%) had full-time jobs and had no work experience (75.3%) in the health care sector. A majority reported having joint replacement surgery (34.3%), spine surgery (17.8%), and wear and tear diseases (11.9%) prior to rehabilitation. Almost half (45.1%) had two or more (chronic) conditions.

A total of 32 HCPs (13 physicians, 6 physiotherapists, 4 social workers, 3 nurses, 6 HCPs without sociodemographic data) were recruited. Across the patient encounters by T0 (*n* = 361), HCPs had to assess several patients. Most HCPs were 50–59 years old (30.8%); 46.2% of HCPs were female (*n* = 12). HCPs graduated an average of 21.7 (SD 13.7) years and worked an average of 8.9 (SD 8.2) years in the current rehabilitation center, with a maximum of 26 years.

### 3.2. Estimations by Patients and HCPs

Figure 1 gives an overview of the estimations by patients and HCPs. Completed T0 (response rate: 99.8%) and T1 (response rate: 79.5%) questionnaires by patients (*n* = 287) were used to describe the level of patient HL. Estimations completed by patients and HCPs (*n* = 278; response rate: 76.8%) were used to analyze HCPs’ and patients’ estimation of patients’ HL and compare the estimation of the patient’s HL by rehabilitation HCPs. The response of the HCPs was as follows: complete data from *n* = 126 estimation by one HCP, *n* = 112 estimation of two HCPs, *n* = 35 estimations by three HCPs, and *n* = 5 estimations by all HCPs were available. Only 3.3% of the patients (*n* = 12) required assistance to complete the questionnaire.

### 3.3. Self-Assessment of Patients’ HL Level

Mean scores for each HLQ scale by the patient (T0 and T1) are displayed in Table 2. For scales 1 to 5, the score range is 1 to 4, and higher scores indicate greater agreement; and for scales 6 to 9, the score range is 1 to 5, with higher scores indicating less difficulty. The mean scores for the four HLQ scales by HCPs were also calculated. Due to the different scoring of the HLQ, of the scales 1–5 (range 1–4), scale 1, “*Feeling understood and supported by health care providers*”, had the highest mean scores (mean = 3.15, SD 0.47 at T0, mean = 3.18, SD 0.46 at T1), and the lowest overall was seen in scale 5, “*Appraisal of health information*” (mean = 2.86, SD 0.48 at T0, mean = 2.89, SD 0.50 at T1), self-assessed by patients. Of the scales 6–9 (range 1–5), scale 6, “*Ability to actively engage with health care providers*”, had the highest mean scores (mean = 3.84, SD 0.57 at T0, mean = 3.92, SD 0.56 at T1), and the lowest mean scores were seen in scale 7, “*Navigating the health care system*” (mean = 3.68, SD 0.56 at T0, mean = 3.79, SD 0.58 at T1), self-assessed by patients.

Regarding the estimation by patients at two points in time (T0 and T1), there was a statistically significant group difference for scale 2, “*Having sufficient information to manage my health*” (*p* < 0.0055, d = −0.35); scale 3, “Actively managing my health”; scale 6, “*Ability to actively engage with health care providers*” (*p* < 0.0055, d = −0.28); scale 7, “*Navigating the health care system*: (*p* < 0.0055, d = −0.14); and scale 8, “*Ability to find good health information*” (*p* < 0.0055, d = −0.18).

### 3.4. HCPs Estimation of Patients’ HL Level

The patients’ self-assessed HL levels (T0) compared with the HCP estimation (T0) can be described as follows: The physicians (total estimations *n* = 176) showed higher mean scores on scale 6, “*Ability to actively engage with health care providers*” (mean = 4.04, SD 0.61), and scale 9, “*Understanding health information well enough to know what to do*” (mean = 4.04, SD 0.58), than the patients. The physiotherapists (total estimations *n* = 141) also presented higher mean scores on scale 6, “*Ability to actively engage with health care providers*” (mean = 4.05, SD 0.52), and scale 9, “*Understanding health information well enough to know what to do*” (mean = 4.30, SD 0.58), than the patients. The social workers (total estimations *n* = 113) demonstrated a higher scale score in three scales (scale 2, “*Having sufficient information to manage my health*” (mean = 2.93, SD 0.61); scale 6, “*Ability to actively engage with health care providers*” (mean = 4.18, SD 0.72); and scale 9, “*Understanding health information well*” (mean = 4.01, SD 0.72) than patients’ self-assessment. Nurses (total estimations *n* = 45) showed a higher mean score in all scales estimated (scale 2, “*Having sufficient information to manage my health*” (mean = 3.24, SD 0.46); scale 3, “*Actively managing my health*” (mean = 3.18, SD 0.45); scale 6, “*Ability to actively engage with health care providers*” (mean = 4.46, SD 0.53); and scale 9, “*Understanding health information well enough to know what to do*” (mean = 4.37, SD 0.57)). Group differences were not tested for significance.

### 3.5. HL Agreement

The intraclass coefficient of agreement was computed to estimate the level of agreement between patient HL levels by patients and HCPs’ predicted level of patient HL levels measured by the HLQ (scale 2, 3, 6, 9).

In identifying specific groups of HCPs, the results demonstrated poor to fair agreement and could be described as follows: The best agreement between HCPs and patients’ overall estimation of patients’ HL level was found at scale 3, and the lowest overall agreement was found at scale 6. The results of scale 3, “*Actively managing my health*”, stated fair agreement between physiotherapists’ (ICC = 0.45), social workers’ (ICC = 0.44), and nurses’ perceptions of patients’ HL and patients’ self-assessment of their HL level.

Physicians’ and patients’ estimations of patients’ HL levels showed fair agreement in scale 9, “*Understanding health information well enough to know what to do*” (ICC = 0.44).

In three scales, social workers and patients had a fair level of agreement (scale 2, “Having sufficient information to manage my health”, ICC = 0.47; scale 3, “Actively managing my health”, ICC = 0.44; and scale 9 “Understanding health information well enough to know what to do”, ICC = 0.56) regarding the estimation of patients’ HL levels. Scale 6, “Ability to actively engage with health care providers” (ICC = 0.31), had poor agreement (see Table 3).

## 4. Discussion

This study aimed to determine inpatient medical rehabilitation patients’ HL based on self-assessment, evaluate changes from admission to discharge, identify rehabilitation HCPs estimation of patients’ HL, and compare the estimated patient HL by patients and HCPs.

Compared to other studies, HL levels at admission were quite similar. A significant increase in HL with small effect sizes was shown in five (scale 2, “*Having sufficient information to manage my health*”; scale 3, “*Actively managing my health*”; scale 6, “*Ability to actively engage with health care providers*”; and scale 7, “*Navigating the health care system*”) out of the nine HLQ scales. These are especially the scales that focused on the rehabilitation process, and these domains improved after three weeks of rehabilitation stay. Whether these changes are caused by interventions, communication, or the fact that patients were in a rehabilitation clinic needs to be researched in further studies. Moreover, these domains (e.g., “*Ability to actively engage with health care providers*”) can be improved in the short term because they do not directly change reading and writing skills (individual skills). Nonetheless, the maintenance of these changes needs to be determined. Other longitudinal studies related to individuals with chronic obstructive pulmonary disease have shown that on scale 3, the main scores decreased after 6 and 12 months [28]. Scales 1, 4, 5, 8, and 9 did not significantly differ between T0 and T1. Although effects do not seem to last for 6 or 12 months, our study shows that there are short-term effects of medical rehabilitation with regard to health literacy. Strategies to perpetuate these effects are needed and should be integrated into after care.

The findings demonstrated higher mean scores estimated by HCPs than patients on most scales. In particular, the mean scores by scale 6, “*Ability to actively engage with health care providers*”, and scale 9, “*Understanding health information well enough to know what to do*”, were higher by all HCPs’ assessment than by patients’ self-assessment. The results of the intraclass coefficient indicated poor to fair agreement between patients’ and HCPs’ estimation of patients’ HL. Our findings for setting rehabilitation are consistent with the previous literature, which showed poor agreement between patients’ and HCPs’ (especially physicians) estimation of patients’ HL [21,29,30,31,32,33,34,35]. HCPs tended to overestimate patients’ HL. Different perspectives between patients and HCPs in estimating the HL of patients may not be because HCPs do not know how they identify patients’ HL levels. After all, they have different perceptions or expectations of HL than patients. For example, scale 6, “*Ability to actively engage with health care providers*”, could be easier for HCPs to estimate, as an impression can be gained through brief communication with the patient. On the other hand, “*Social support for health*” (scale 4) may be more difficult to estimate by HCPs in rehabilitation, as they do not have an accurate overview of the social background of patients in rehabilitation. 

Social workers had fair agreement on three out of four scales, and all other HCPs had fair agreement on only one scale. All other estimations were in poor agreement. Reasons for a better agreement but lower scale scores could be that the social workers had focused contact, which enables enough time to identify and adapt to patients’ communication needs during the initial interview. A quiet conversation environment and as few sources of irritation as possible are helpful in this context. Different estimation results between the HCPs could also be due to different training statuses and working experiences [36,37]. The different assessments of HCPs for individual patients should also be discussed within the team, so it is also useful to improve communication within a caring team.

To enhance patients’ disease knowledge, their ability to self-manage, and adherence to health care recommendations, awareness of the importance of HL may be integrated into clinical practice by the HCP rehabilitation team [38,39]. HCPs should intervene to improve HL because patients with chronic disease often use rehabilitation programs and can benefit from them.

### 4.1. Implications for Practice and Research

Differences in HL estimation in patients and HCPs may lead to communication problems. On-the-job training for HCPs could be useful [40]. This training should focus on identifying the needs of patients with low HL and include strategies to promote communication between HCPs and patients. For example, the “teach-back” [41] or “chunk and check” [42] methods can be used. With these methods [41,42], HCPs check patients’ understanding by asking them to reproduce in their own words what was explained to them by the health care professional [43]. HL communication training for HCPs in hospital settings was developed and successfully tested in several countries and diverse settings [17,18,44,45]. The results showed that this training subjectively improved HCP knowledge about HL, understanding HL needs, awareness of their jargon, self-efficacy, and adaptations in patient interactions [44,46]. Other effects of the differences in HL estimation in patients and HCPs should be addressed in further research.

### 4.2. Strengths and Limitations

A strength of the study is that a multidimensional instrument, the HLQ [23], was used to assess patients’ HL according to patients and HCPs to reflect different domains of HL. In addition, the research questions were answered through the participation of different HCP groups in an unresearched setting, the rehabilitation setting. In contrast, previous studies [18] have provided new findings on different professional groups. In addition, there were high response rates among patients (response rates T0: 99.8%; T1: 79.5%) and HCPs (overall response rate: 76.8%). However, we conducted the study only in one center focused on orthopedic and rheumatological rehabilitation. Therefore, a homogeneous patient group can be assumed, which does not reflect the diversity of patients in rehabilitation. Another limitation of the study is that only a small number of HCPs was recruited, posing a risk of identification. Additionally, no exact number of patients estimated per HCP can be given because HL estimation of HCPs was anonymized. Furthermore, to avoid multiple testing, data of the comparison of patients (T0) and HCP HLQ scale scores were not tested for statistical significance.

## 5. Conclusions

This study provides preliminary evidence on the patients’ and HCPs’ estimation of rehabilitation patients’ HL and their agreement. The results reveal that patients’ and HCPs’ estimation of patients’ HL often dissents. Communication training could help to improve communication between patients and HCPs. Rehabilitation is particularly linked to HL [39], and the topic of HL should be given higher awareness in the rehabilitation process to improve patients’ HL, including patients with chronic diseases.

## Figures and Tables

**Figure 1 ijerph-19-03522-f001:**
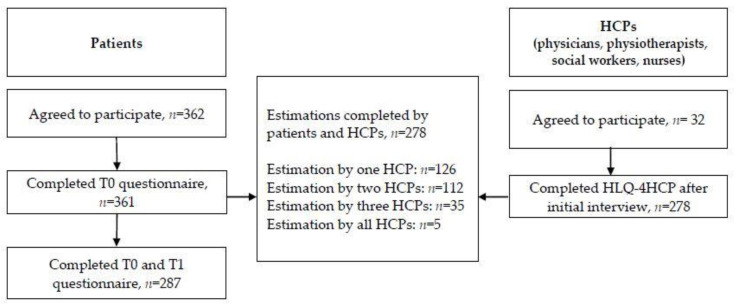
Estimations by patients and HCPs.

**Table 1 ijerph-19-03522-t001:** Demographic characteristics of patients.

Demographic and Health Characteristics	Total *n* = 287
Age, *n* (%)	
Until 29 years	4 (1.4)
30–39 years	15 (5.2)
40–49 years	32 (11.1)
50–59 years	140 (48.8)
60 years and older	95 (33.1)
Prefer not to answer	1 (0.3)
Gender, *n* (%)	
Women	178 (62.0)
Men	109 (38.0)
Live alone, *n* (%)	
Alone	55 (19.2)
Not alone	224 (78.0)
Prefer not to answer	8 (2.8)
Nationality, *n* (%)	
German	273 (95.1)
Other	6 (2.1)
Prefer not to answer	8 (2.8)
Mother tongue, *n* (%)	
German	261 (90.9)
Other languages	17 (5.9)
Prefer not to answer	9 (3.1)
Language skills in German	
very good	3 (17.6)
good	12 (70.6)
medium	2 (11.8)
Education, *n* (%)	
A-level or higher	120 (41.8)
GCSE or equivalent	153 (53.3)
No Qualification	5 (1.7)
Prefer not to answer	9 (3.1)
Reasons for rehabilitation, *n* (%)	
Joint replacement surgery (e.g., hip, knee)	135 (34.3)
Spine surgery (e.g., intervertebral disc)	70 (17.8)
Wear disease (e.g., arthrosis)	47 (11.9)
Functional disorders of the musculoskeletal system	46 (11.7)
Chronic back pain	38 (9.7)
Bone fracture, muscle-tendon rupture, capsular ligament injury	21 (5.4)
Other diseases	35 (9.0)
1 (chronic) disease	215 (54.9)
≥2 (chronic) diseases	177 (45.1)
Status of disability	
No severe disability	250 (87.1)
Severe disability (≥50%)	35 (12.2)
Not sure	2 (0.7)

**Table 2 ijerph-19-03522-t002:** Patients (T0, T1)—Health Literacy Questionnaire (HLQ) scale scores and *t*-tests for dependent samples; HCPs (T0)—Health Literacy Questionnaire (HLQ) scale scores.

HLQ Scale	Mean (SD) (95% CI), Patients T0 (*n* = 287) ^a^	Mean (SD) (95% CI), Patients T1 (*n* = 287) ^b^	*p*-Value *	Cohens *d ***	Mean (SD) (95% CI), Physicians (*n* = 176)	Mean (SD) (95% CI), Physio- Therapists (*n* = 141)	Mean (SD) (95% CI), Social Workers (*n* = 113) ^c^	Mean (SD) (95% CI]), Nurses (*n* = 45) ^d^
Range 1 (lowest)–4 (highest)								
1. Feeling understood and supported by health care providers	3.15 (0.47) (3.09, 3.20)	3.18 (0.46) (3.12, 3.23)	0.261					
2. Having sufficient information to manage my health	2.85 (0.50) (2.80, 2.91)	3.02 (0.48) (2.96, 3.08)	0.000	−0.35	2.82 (0.61) (2.73, 2.91)	2.73 (0.60) (2.63, 2.83)	2.93 (0.61) (2.82, 3.05)	3.24 (0.46) (3.09, 3.38)
3. Actively managing my health	2.95 (0.44) (2.90, 3.00)	3.07 (0.42) (3.02, 3.12)	0.000	−0.28	2.73 (0.69) (2.63, 2.83)	2.90 (0.54) (2.81, 2.99)	2.76 (0.54) (2.66, 2.86)	3.18 (0.45) (3.03, 3.32)
4. Social support for health	3.06 (0.50) (3.00, 3.11)	3.10 (0.49) (3.04, 3.15)	0.082					
5. Appraisal of health information	2.86 (0.48) (2.80, 2.91)	2.89 (0.50) (2.83, 2.95)	0.113					
Range 1 (lowest)–5 (highest)								
6. Ability to actively engage with health care providers	3.84 (0.57) (3.77, 3.90)	3.92 (0.56) (3.86, 3.99)	0.003	−0.14	4.04 (0.61) (3.95, 4.13)	4.05 (0.52) (3.96, 4.13)	4.18 (0.72) (4.04, 4.31)	4.46 (0.53) (4.30, 4.62)
7. Navigating the health care system	3.68 (0.56) (3.62, 3.75)	3.79 (0.58) (3.72, 3.86)	0.000	−0.19				
8. Ability to find good health information	3.79 (0.56) (3.72, 3.85)	3.89 (0.55) (3.82, 3.95)	0.000	−0.18				
9. Understanding health information well enough to know what to do	3.83 (0.55) (3.77, 3.89)	3.89 (0.55) (3.83, 3.96)	0.011		4.04 (0.58) (3.95, 4.13)	4.30 (0.58) (4.20, 4.40)	4.01 (0.72) (3.87, 4.14)	4.37 (0.57) (4.20, 4.54)

^a^ Missing data, patients T0: Scale 1 = 1, scale 6 = 1, scale 7 = 1, scale 8 = 1, scale 9 = 1; ^b^ Missing data, patients T1: Scale 2 = 1, scale 5 = 2, scale 6 = 3, scale 7 = 4, scale 8 = 3, scale 9 = 3. ^c^ Missing data, Social workers: Scale 3 = 1; ^d^ Missing data, nurses: Scale 2 = 4, scale 3 = 4, scale 6 = 1, scale 9 = 1. * Bonferroni correction: *p* < 0.0055; ** Cohen (1988): <0.5—small effect size; 0.5–0.8—medium effect size; > 0.8—large effect size.

**Table 3 ijerph-19-03522-t003:** Intraclass coefficient (ICC) of HCPs’ and patients’ estimation of patients’ HL.

		ICC *			
Patients	**HLQ Scale**	**Physicians** (***n* = 176**)	**Physiotherapists** (***n* = 141**)	**Social Workers** (***n* = 113**)	**Nurses** (***n* = 45**)
	2. Having sufficient information to manage my health	0.21	0.20	0.47	0.10
	3. Actively managing my health	0.27	0.45	0.44	0.40
	6. Ability to actively engage with health care providers	0.21	0.02	0.31	0.16
	9. Understanding health information well enough to know what to do	0.44	0.16	0.56	0.27

* Cicchetti (1994): Less than 0.40—poor agreement; between 0.40 and 0.59—fair agreement; between 0.60 and 0.74—good agreement; between 0.75 and 1.00—excellent agreement.

## Data Availability

The data presented in this study are available on request from the corresponding author.

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
