# Peer review of "Patients’ Health Literacy in Rehabilitation: Comparison between the Estimation of Patients and Health Care Professionals"

_ijerph, 2022, doi:10.3390/ijerph19063522_

Round 1
Reviewer 1 Report
This is a well-designed study which revealed poor agreement between patients and professionals’ assessment of patients’ health literacy. I have only one suggestion that authors might consider. In my opinion exclusion criteria of patients should be described in details. I guess that patients with some disorders and not only patients with language difficulties were excluded from the study (for example patients with psychiatric or psychological or cognitive disorders).
Reviewer 2 Report
The paper is very interesting especially because of the opportunity to compare patient and HCPs perspectives. However, some things need to be clarified.
- The most unclear issue is the use of tests for paired data. First, the authors use a colloquial term (connected and disconnected samples - line 170). Better would be dependent and independent samples. The comparison of T0 with HCP opinion is on much smaller samples, so patient means are not the same as in a total sample (N=287) For this comparison, significance is not given at all (lines237+), only M+/-SD from the table for HCP is quoted. It would be optimal to include the tables in the supplementary appendix (or 4 tables for comparisons of patients with 4 groups of HCPs). If for some reason data were not linked then this should be stated in the limitations (lack of paired comparisons).
- It might also be a good solution to split Table 3 into two separate.
- In the methodology, it is worth noting that (if true) each patient was assessed once by a staff member from a given group (and not, for example twice by two physicians in the absence of data pairing).
- The first objective (line 71 and abstract) omits the longitudinal part of the study.
- In the description of the tool, it is worth noting whether it is publicly available. There is only a reference to item [22]. Sample questions would be interesting when presenting the HL dimensions (lines 98+).
- The longitudinal section is poorly described. T1 was defined as 5 days before discharge. But the time between T0 and T1 is not shown. This would help a lot in interpreting the results, because the discussion mentions a comparison with 6 and 12 months later. Also, there is no hypothesis in the discussion or introduction as to why it might improve during rehabilitation treatment. Was any attention paid to improving patient->HCP communication as part of the intervention during treatment?
- One of the most important practical conclusions is the need to improve patient->HCP communication. I think this issue should also be hinted at in the introduction as well. Here the authors get into the fascinating topic of humanization in medicine.
- There is an interesting result not discussed in the text, that nurses score higher on most dimensions, but this is associated with lower level of agreement. In the case of social workers, agreement is the best but not as high ratings. Drawing attention to social workers as an important occupational group is a major strength of this work.
- The abstract needs rewriting. It has overly elaborate objectives, a poorly described tool, and few specific outcomes. Two sentences (lines 21 and 22) are in some contradiction. Higher ratings by employees, but weak correlation. Also no reference to T0 and T1 comparisons.
- Line 326 is HLC and not HCP, is it Ok?
- The discussion is carefully written, although relatively short. A couple of points would be worth emphasizing. First, HL can be by definition and in some areas a subjective assessment (like quality of life) and HCPs may not have objective knowledge of the patient, they are just giving their observations. Second, the results regarding difference by patient HL as perceived by different groups of HCPs suggest that HCPs should discuss that topic. Means it would also be worth improving communication within the team caring for a particular patient.
- Please check the numbering of the tables, where is table 2?
Reviewer 3 Report
The paper provides an interesting insight into the agreement between healthcare personnel and patient in the assessment of patient’s health literacy in a rehabilitation setting.
Data are clear, the methodology is correct - a few suggestions to improve the manuscript.
Introduction
Ref. number 3 is outdated. A more recent meta-analysis estimated the prevalence of low health literacy in European countries, including Germany (doi: 10.1007/s11606-020-06407-8).
Methods
It’s unclear why the authors asked patients to assess their health literacy level twice (T0 and T1). A brief explanation should be provided.
Results
Which was the response rate for patients? And for the HCP?
Discussion
The fact that the mean self-reported health literacy levels were higher at T1 than T0 could be due to the fact that these patients were exposed to a healthcare environment during rehabilitation, therefore it could be the reason why the felt more confident about their health literacy. This or alternative explanations should be discussed.
Findings indicate poor agreement between patients and HCPs estimation of patients’ health literacy. What was the impact of that discrepancy? The discussion should elaborate on this.
How you chose to measure health literacy (the measurement tool) and the implications for your findings should be added. Good references are doi: 10.1080/10810730.2013.829892 and doi: 10.1186/1471-2458-14-1207.
Limitations
Response rate is not mentioned but should be addressed.
Round 2
Reviewer 2 Report
Congratulations and I have no further comments.
Author Response
Dear editors,
we are very happy to point us to our mistake with regard to IP. We did not do this on purpose, it was an oversight. Thank you very much for your suggestions for the further process. We now have an approval for our adaptations for administering the HLQ to HCPs.
With kindest regards,
Mona Voigt-Barbarowicz (on behalf of all authors)
